# Deep Q-Learning Technique for Offloading Offline/Online Computation in Blockchain-Enabled Green IoT-Edge Scenarios

Arash Heidari [1], Mohammad Ali Jabraeil Jamali [1,*], Nima Jafari Navimipour [2,3,*] and Shahin Akbarpour [1]

1 Department of Computer Engineering, Shabestar Branch, Islamic Azad University, Shabestar 5381637181, Iran
2 Department of Computer Engineering, Tabriz Branch, Islamic Azad University, Tabriz 5157944533, Iran
3 Department of Computer Engineering, Faculty of Engineering and Natural Sciences, Kadir Has University, Istanbul 34083, Turkey
* Correspondence: m_jamali@itrc.ac.ir (M.A.J.J.); nima.navimipour@khas.edu.tr (N.J.N.)

**Abstract:** The number of Internet of Things (IoT)-related innovations has recently increased exponentially, with numerous IoT objects being invented one after the other. Where and how many resources can be transferred to carry out tasks or applications is known as computation offloading. Transferring resource-intensive computational tasks to a different external device in the network, such as a cloud, fog, or edge platform, is the strategy used in the IoT environment. Besides, offloading is one of the key technological enablers of the IoT, as it helps overcome the resource limitations of individual objects. One of the major shortcomings of previous research is the lack of an integrated offloading framework that can operate in an offline/online environment while preserving security. This paper offers a new deep Q-learning approach to address the IoT-edge offloading enabled blockchain problem using the Markov Decision Process (MDP). There is a substantial gap in the secure online/offline offloading systems in terms of security, and no work has been published in this arena thus far. This system can be used online and offline while maintaining privacy and security. The proposed method employs the Post Decision State (PDS) mechanism in online mode. Additionally, we integrate edge/cloud platforms into IoT blockchain-enabled networks to encourage the computational potential of IoT devices. This system can enable safe and secure cloud/edge/IoT offloading by employing blockchain. In this system, the master controller, offloading decision, block size, and processing nodes may be dynamically chosen and changed to reduce device energy consumption and cost. TensorFlow and Cooja's simulation results demonstrated that the method could dramatically boost system efficiency relative to existing schemes. The findings showed that the method beats four benchmarks in terms of cost by 6.6%, computational overhead by 7.1%, energy use by 7.9%, task failure rate by 6.2%, and latency by 5.5% on average.

**Keywords:** Blockchain; deep learning; IoT; Offloading; QoS; privacy

## 1. Introduction

The growth of the Internet of Things (IoT)-enabled services, including industrial automation, radar sensor data, e-healthcare, smart cities, vehicle routing control, smart homes, and so on, has led to a massive rise in the volume of data and allows access to information about the physical world [1–6]. Because of the excessive separation between IoT objects and the cloud server, the traditional two-layered cloud-based IoT design suffers from significant data processing delays, making it unsuitable for delay-sensitive apps [7,8]. To address this limitation and meet the demands of delay-sensitive applications, a three-layered IoT–edge–cloud structure has received a lot of attention. The edge platform provides cloud services at the network's edge to minimize the latency of processing activities performed by terminal nodes [9]. Edge computing decreases latency and the quantity of data transmitted to the cloud for processing, making it ideal for delay-sensitive IoT apps. This massive expansion necessitates systems to handle the growing number of IoT objects as

well as create and communicate the generated mobile data [10,11]. The devices' low power and processing capabilities, such as CPU and RAM, significantly limit their ability to run such resource-intensive apps [12]. To improve Quality of Service (QoS), particularly for IoT latency-sensitive applications, computation offloading is a strategy that transfers compute-intensive jobs or applications from the resource-constrained IoT devices to more processing-capable devices such as the cloud or edge. Therefore, delay-sensitive computations could be offloaded to more innovative platforms to overcome these restrictions and fulfill the communication/processing latency requirement.

Additionally, Energy Harvesting (EH) is a promising way of extending the battery's life and supplying objects in a wide range of applications [13–16]. IoT objects using EH components must resist eavesdroppers that analyze sensing data over radio channels to reveal the user's position and activities, such as the privacy of the use pattern [17–20]. Therefore, EH techniques can be used by IoT systems to use environmental energy, such as body motion and ambient Radio Frequency (RF), to prolong battery life [21]. Furthermore, privacy is vital in offloading systems. Imagine the IoT system offloads all of the sensing data to the edge device while the radio channel is in excellent shape [22]. In such an instance, an attacker might identify the usage pattern by estimating the amount of the newly produced sensing data [23,24]. Consequently, the IoT system must cover user location and usage pattern privacy in device offloading. Blockchain is used to initially record and store transactions as an underlying Bitcoin technology [25,26]. Compared to a traditional centralized ledger managed by a third party, a blockchain is a distributed ledger with a Peer to Peer (P2P) network structure [27]. Therefore, it will guarantee the confidentiality and authenticity of data effectively. Regardless of whether it is a permissionless blockchain like Bitcoin or Ethereum, or an authorized blockchain like Byzantine Fault Tolerance (BFT), they all require a lot of mining or consensus computational resources [28]. Therefore, the reliability of blockchain computing and security are important concerns in the IoT [25,29].

This research aims to create a low-energy-consumed short-delay computing system by combining IoT-edge, EH module, a blockchain platform, and Deep Learning (DL) technique [30]. To optimize latency, energy consumption, and battery life, we offer a novel version of the reinforcement learning approach to the Markov Decision Process (MDP) problem, making the most of the optimal decisions for online/offline offloading. When the state space is large, the MDP suffers from the well-known "curse of dimensionality" issue. For this reason, to execute an optimum offloading policy as a learning agent for an IoT system, we propose a Dynamic Online/Offline Q-learning-based approach for IoT-Edge Offloading called DO$^2$QIEO. In addition, we use a Post-Decision State (PDS)-based learning approach to learn the best joint offloading for online scenarios. To overcome this problem, the PDS-based learning algorithm uses the unique structure of the considered EH's state transitions, greatly boosting run-time performance and learning convergence speed. Hence, we utilize the EH module to extend the life of the battery. The current state and the Q-function value are used to determine the offloading policy, including the edge device and the offloading rate of IoT objects to determine the incentive or efficiency benefit based on energy consumption, task failure rate, and computation delay within, and time slot. This procedure, in turn, generates the most recent policy by updating the Q-function, the efficiency-based iterative Bellman equation in offline mode, and the PDS in online mode. Once the system is online, the circumstance must be changed. An online deep Q-learning algorithm has been presented to deliver the best policy on the fly. To expedite learning, the PDS method employs partially known knowledge about dynamic systems and allows the edge node to incorporate such knowledge into its learning experience. The DO$^2$QIEO method is based on the PDS, which can handle dynamic offloading in real-time. Thus, to enhance long-term offloading performance, we use our DL method to handle highly dynamic settings while also addressing computing complexity. We use Transfer Learning (TL) during exploration to accelerate convergence by minimizing unnecessary exploration. As a result, the TL is utilized in the technique to launch the system. Effective TL is required to maximize learning outcomes. Furthermore, some of the primary benefits include reduced

training data, improved model generalization, and increased DL usability. Further, because the system can transition from offline to online, it could be used in real-world scenarios. The key contributions of this study are listed below.

- Proposing a secure framework for integrating edge and blockchain technologies into IoT networks to ensure data protection and energy efficiency;
- Taking into account the computing servers and the status of the controllers, we design the optimization problem as an MDP by specifying state, action, and reward function simultaneously, as well as the dynamic features of IoT systems;
- Providing a platform for dynamic online/offline offloading for various IoT-edge applications called DO$^2$QIEO;
- Using the TL method can assist in achieving the best offloading strategy;
- Using the EH module to increase the battery's life and improve offloading performance;
- Increasing system performance by lowering energy consumption, reducing computational latency, increasing device efficiency, and minimizing the task failure rate.

The rest of this paper is structured in the following manner. Section 2 deals with related work. Section 3 describes the system model and problem statements. Section 4 presents the proposed method. Section 5 deals with performance evaluation. Section 6 discusses the conclusions and future scope.

## 2. Related Work

The development of IoT offloading methods is a hot topic of study, and a lot of work has been done in this field. Therefore, depending on their characteristics, advantages, and weaknesses, we have listed the selected related works in this portion.

Several Q-learning spin-off techniques are used in this field. Aljanabi and Chalechale [3] suggested a task offloading approach to a particular fog or the cloud node to establish the best decision on where and when to offload a task. As an MDP, the topic is provided and analyzed. Two decision-makers were discussed in their suggested MDP, where IoT objects may choose the fog platform to which they need to offload their tasks, while fog servers can decide to offload those jobs to other fog or to cloud nodes to balance the tasks. A Q-learning-based model is applied to achieve the optimal policy to manage large-size state and action space. The findings showed that their technique achieves improved load balancing and decreases the latency relative to other works.

Fog computing allows IoT systems to enable applications insensitive to latency and computational intensity with lower overhead in computation and energy consumption. Therefore, Ramanathan, Williams [31] investigated fog-enabled IoT-eHealth networks' offloading and migration problems. First, they described a total time allocation model that considers transmission, migrations, calculation, and a penalty for unreasonable offloading decisions. Thus, multi-stage stochastic programming was utilized to develop the offloading issue as a stochastic optimization to determine the effect of ambiguity on the offloading strategy in stochastic settings. Besides, they examined the efficiency of stochastic multi-stage programming, considering the limited potential of energy and computation. The results revealed that their method's performance can be improved relative to baseline systems, while their technique can lower total computational complexity.

Additionally, privacy is crucial for IoT-fog computing schemes, and several strategies are applied to various offloading techniques that make the network reliable and powerful, such as blockchain, etc. To build a node access allocation policy based on the types of health information, Zhang, Cho [32] suggested a simulated annealing-based offloading algorithm. In this manner, the quickest response can be obtained by users. Certain individuals may eventually be recognized to suffer emergency injuries, which means these situations must be managed with the highest priority; otherwise, life threats may be posed. The findings showed that the algorithm for emergency measures would efficiently help patients find the best way to connect with competent medical units. Their strategy primarily evaluated offloading traffic to reduce latency in the healthcare system.

Several DL spin-off techniques are used in fog/edge computing and industrial IoT networks for offloading. Yuan, Tian [33] suggested a Two-Stage Possible Game-based Computation Offloading strategy (TPOS) for Wireless Body Area Networks (WBANs). They divided the game space into two stages to solve the multi-user game problem. Initially, jobs with various priorities in WBANs were assigned offloading choices based on their utility and penalty functions. The MEC node allocated computational resources on the second level to offload tasks via the potential game. Even in heavy tasks and dense WBAN scenarios, their research revealed that the TPOS algorithm could satisfy low latency and low energy consumption. The estimate of various forms of uncertainty is required to create active learning paradigms, novelty detection, error detection, and data fusion, all of which are critical for developing successful and tailored activity recognition systems.

Besides, Wu, Wolter [34] highlighted significant offloading difficulties in blockchain-enabled heterogeneity IoT-edge-cloud computing settings, where edge platforms can collaborate to reduce IoT object energy consumption while meeting delay limitations. They presented the offloading issue as an optimization problem. They developed a polynomial-time-complexity method using the Lyapunov optimization method, which evaluates if and where to offload to reduce the IoT device's energy consumption. They also demonstrated that the cloud could be used as a long-term data processor for computation-intensive jobs, whereas simulations confirmed the solution's efficacy. Validating real-world blockchain networks with smart agents was required.

Finally, Zuo, Jin [35] suggested a cooperative edge-assisted blockchain network, in which Proof-of-Work (PoW) mining operations might be delegated to Base Stations (BSs), and block data could be kept by a Cloud Service Provider (CSP). They used a 3-stage Stackelberg game to model their network's interaction process between IoT objects, BSs, and CSP. Within every step, they also looked at the subgame optimization problem. They developed the Stackelberg game's backward induction-based iterative approach for computing offloading, block storage strategies, and resource service prices for all equipment, BSs, and CSPs. They found that their proposed backward induction-based iterative method had a high convergence rate and discussed the interactions between the three stages of the proposed game. In addition, the analysis results revealed that the two-cell method had significant advantages over the two non-cooperative methods. Finally, a side-by-side analysis of state-of-the-art IoT offloading methods is shown in Table 1.

**Table 1.** A comparison of the discussed IoT offloading methods and their characteristics.

| Mechanism | Main Idea | Parameters | | | | | | Network | Using TL? |
|---|---|---|---|---|---|---|---|---|---|
| | | Convergency | Stability | Energy | Latency | Task Failure | Security | | |
| Aljanabi and Chalechale [3] | Proposing a Q-learning-based method to fix the model and select the optimal offload policy. | ● | ● | ● | ● | ● | ● | Fog-cloud | No |
| Ramanathan, Williams [31] | Dealing with the uncertainty of link time using a multi-stage offloading method. | ● | ● | ● | ✓ | ● | ● | Fog-IoT | No |
| Zhang, Cho [32] | Suggesting an offloading scheme to create a node access policy. | ● | ● | ● | ✓ | ● | ● | Edge-cloud | No |
| Yuan, Tian [33] | Suggesting a two-stage game-based offloading method. | ● | ● | ✓ | ✓ | ● | ● | IoT-MEC | No |
| Wu, Wolter [34] | Suggesting an online offloading based on the Lyapunov optimization. | ● | ● | ● | ● | ● | ● | IoT-edge-cloud | No |
| Zuo, Jin [35] | Proposing a three-stage Stackelberg game for offloading together | ● | ● | ● | ● | ● | ● | Mobile-edge computing | No |
| Our work | Proposing a DL method for online/offline offloading enabled blockchain. | ● | ● | ✓ | ✓ | ✓ | ✓ | IoT-edge-cloud | Yes |

Using Deep Neural Network (DNN) approaches on IoT objects can result in overhead; plus, finding ways to remove this overhead or lighten these algorithms on more capable devices, such as edge nodes, can help to overcome these flaws. Additionally, considering security and privacy mechanisms, blockchain technology can improve the security of a system. Moreover, we are using an EH module to extend the system's battery life. However, we are attempting to build an intelligent and online/offline dynamic approach for secure and cost-effective task offloading that outperforms the smart and dynamic methods, employing the most up-to-date techniques and expertise.

## 3. System Model and Problem Statements

In this part, we introduce the system's model and formulate the problem.

### 3.1. System Model

Offloading is useful for overcoming potential IoT constraints such as low battery power and bandwidth [36]. Offloading is defined as determining which computing tasks should be offloaded and how limited resources should be planned to ensure QoS metrics. Furthermore, the offloading decision should take into account the user's QoS needs, such as processing delay, energy usage limits, and so on. Figure 1 shows a three-layer structure for IoT task offloading. The edge computing structure is viewed as an interlayer between the two. This structure is critical for increasing data offloading privacy while processing tasks quickly and returning them to the IoT system. In this system, IoT delay-sensitive tasks can be split into equal pieces and offloaded to other layers. Machine Learning (ML) components are integrated into the objects and the edge nodes [37,38]. Nevertheless, thanks to evolving edge device technology, the combination of blockchain and edge allow for secure network and computation access and control at the edges. Since both blockchain and edge have a distributed structure, they are better suited to each other. The incorporation of edge and blockchain into IoT is advantageous, given the high-security requirements and intensive computation in IoT networks. The proposed system model is divided into three layers, as shown in Figure 2: IoT layer, edge layer, and blockchain. These three layers' models are depicted and portrayed in the following manner. The list of symbols used in the paper is provided in Table 2.

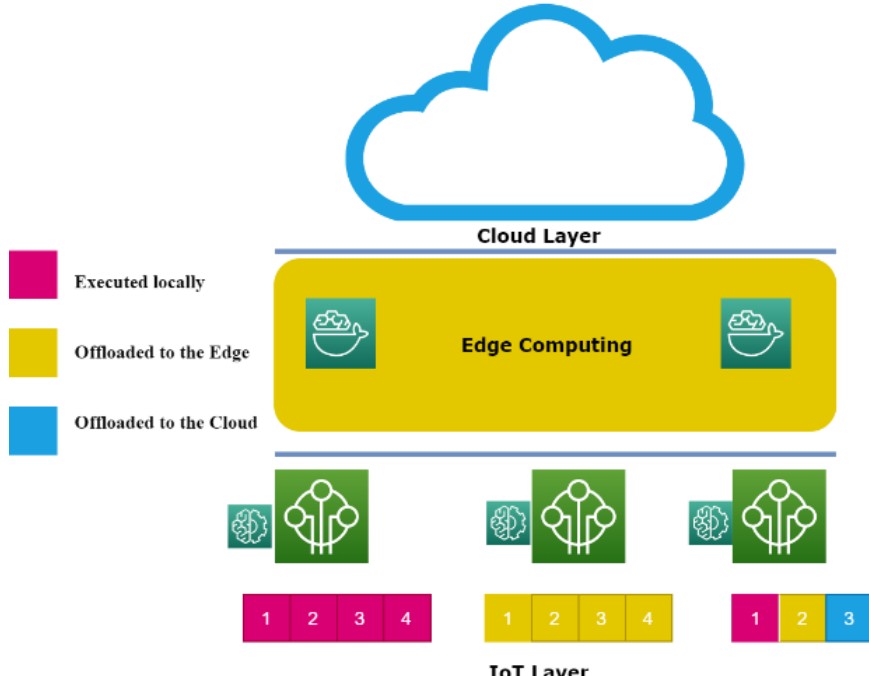

**Figure 1.** Using multiple layers and networks to offload different aspects of tasks.

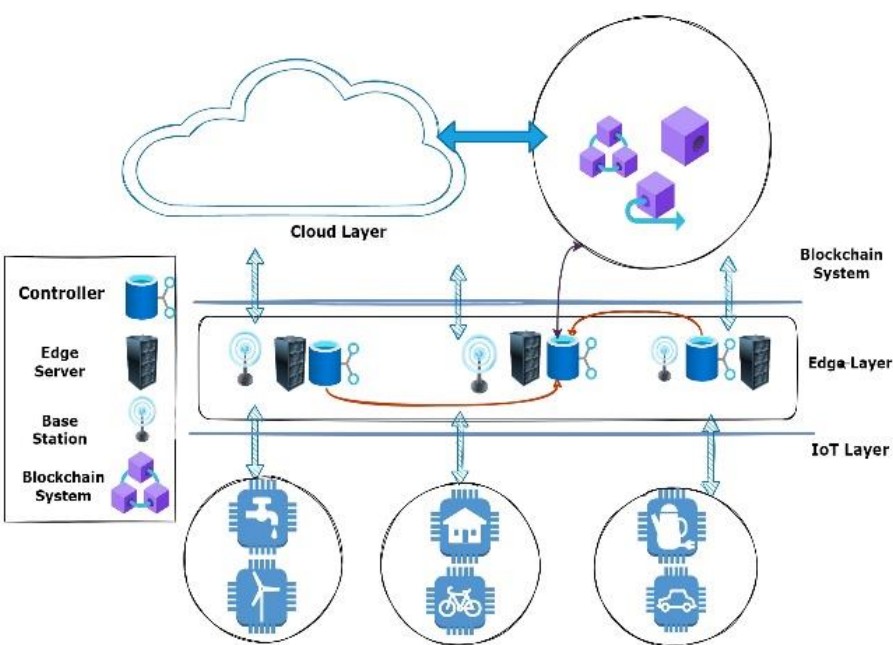

**Figure 2.** The DO²QIEO network scenario's architecture, which includes a cloud, edge, and blockchain system.

**Table 2.** Notations List.

| Symbol | Description |
|---|---|
| $O$ | The number of IoT devices |
| $E$ | The number of edge devices |
| $\mathbb{T}$ | Time slots |
| $n$ | The job of IoT devices |
| $A$ | Set of network tasks |
| $p$ | IoT object's attributes |
| $w$ | Task attributes |
| $g$ | Network attribute |
| $\beta$ | Harvested energy |
| $b$ | Battery level |
| $f$ | Frequency of CPU clocks |
| $Z$ | The set of cells |
| $Z''$ | Group of controllers |
| $d_i$ | The size of the computation task |
| $d_j$ | The delay provisions of the job |
| $CE$ | Contains the energy of all controllers |
| $cc$ | Computation's overhead |
| $CC(t)$ | The total computing cycles |
| ή | Computation resources |
| $E_{local}$ | Local energy consumption |
| $E_{offload}$ | The energy consumption of data transmission |
| $E_{tocal}(t)$ | Energy consumption of Total execution |
| $E_{blockchain}(t)$ | Energy consumption of blockchain system |
| $q_t$ | The computational complexity of the tasks |
| $\beta_z$ | Energy Consumption Coefficient |
| $pt$ | The MC's transmission capacity |
| $bn$ | The number of blockchain nodes |

**Table 2.** *Cont.*

| Symbol | Description |
|--------|-------------|
| $c_c$ | The controller $z$'s computing capability |
| $tr_{cd}$ | Rate of transmission between the controllers and the local edge server |
| $tr_{cb}$ | The transmission rate between the controllers and the blockchain |
| $CO_{offload}$ | The computation overhead |
| $CO_{server}$ | Computation overhead by server |
| $CO_{total}$ | Total overhead |
| $CO_{block}$ | Block overhead |
| $M$ | The number of consensus nodes |
| $cc_1$ | Request period's computing cycles |
| $cc_m$ | The necessary computing cycles at the MC |
| $l$ | The percentage of right transactions sent by the MC |
| $tb$ | The transaction batch's total size. |
| $ts$ | The average transaction size |
| $Cost$ | System cost |
| $j$ | The block size |
| $\partial$ | The constant-coefficient |
| $b_t$ | The broadcast delay among nodes |
| $ti$ | The block generation interval |
| $V_n(t)$ | Consensus time |
| $Ef_t$ | The IoT object's efficiency |
| $C$ | The cumulative gain of the IoT antenna |

*3.2. Problem Formulation*

In this section, we formulate the system with MDP to explain the problem, intending to reduce the system weighted cost based on the specifications of large-dimensional and high-dynamic blockchain-enabled IoT networks. The IoT object, which could be a camera or a utility object, is attached to an energy harvester module, which must handle computation-intensive tasks such as augmented reality apps. These jobs must be completed promptly to provide high customer satisfaction and direct subsequent actions locally or remotely. Additionally, our approach uses local edge nodes made up of basic computers and laptops.

Besides, EH components and electricity storage elements, including RF energy harvester, are included in the IoT object. The IoT system could either perform jobs locally or offload jobs to an edge platform that could offer high computation efficiency with a virtual machine, thanks to the energy provided by the EH element. Typically, IoT objects are the slowest in terms of processing speed, and the cloud is particularly quick in terms of preparing; on average, IoT processing power < edge processing power < cloud processing power. As a result, IoT can connect to the cloud via Wide Area Network (WAN) and the edge via ZigBee and Bluetooth at varying powerful transmission rates. Thus, the edge platform and the IoT object can rapidly acquire the properties of the task.

In this system, the number of edge nodes represents by $E$ and the number of IoT objects represents by $O$, which can quickly obtain the properties of a network task. Additionally, $t \in T$ is the set of slots of all time. Job $n \in A$ can be measured in this system by the data size $d_i$. IoT apps are made up of computational jobs, all of which can be offloaded. Additionally, each job can be broken down into multiple identical parts, and the computation depends on the number of instructions. The object state for IoT systems and tasks is specified as *state* $s = (p, w, g)$, where $p$ is the IoT object's attributes, $w$ is the attributes of the task, and $g$ contains networks status. In a nutshell, $p = (\beta(t); b; f)$, where $\beta$, $b$, and $f$ are harvested energy, battery level, and frequency of CPU clocks, respectively. Additionally, $w = (d_i; d_j)$, where $d_i$, and $d_j$ are the data size and the delay provisions of the job. The $g = (CE, CC, \acute{\eta})$ contains the energy of all controllers, overhead computation, and

computation resources, respectively. To simplify, we split spaces within discrete cells by grid approach and configure delay provisions in multiple domains.

In this system, the IoT controller will pick one of the actions to be taken to deal with the task at each decision time. The controller module is part of an IoT object, and it has the deep method, which allows the object to select whether to process data locally or offload it to an edge node. The actions can be viewed using the set $\hat{A}$ = $\{a_{MC}, a_{Off}, a_{BS}, a_{Ss}\}$, where $a_{MC}$, $a_{Off}$, $a_{BS}$, and $a_{Ss}$ indicate that the Master Controller (MC) selection, the offloading decision, the block size, and the service selection, respectively. Where $a_{MC}(t) \in (1, 2, 3, \ldots, U)$ which controller is elected to be the MC. Additionally, $a_{Off} \in (0, 1)$ refers to the task offloading decision, with $a_{Off} = 0$ denoting that the IoT object performs the jobs locally and a $a_{Off} = 1$ denoting that the MC offloads the jobs. In addition, $a_{BS}(t) \in (1, 2, 3, \ldots, x)$ is the block size level, and $a_{SS}(t) \in (1, 2, 3, \ldots, Q)$ means which node is chosen to support the consensus process of nodes.

### 3.2.1. The IoT Layer

As shown in Figure 2, the system is made up of various IoT cells, and $E$ edge nodes are considered. Moreover, the functions of IoT objects should be completed on time to provide a positive user experience and to direct subsequent actions taken remotely or locally. The IoT object includes EH modules and electricity storage elements such as RF energy harvesters and photovoltaic modules. A significant number of IoT objects are deployed and connected to the local controller in each cell. These IoT objects collect relevant local data during working hours and regularly offload the collected data to the local controller. IoT objects, for instance, regularly send power consumption [38] data to their local controller. Additionally, there is an MC assigned to gather the different IoT data from all local controllers at each time slot $t$. The MC then processes and packages the data before being sent to the blockchain system for data consensus and recording.

### 3.2.2. Edge Layer

There is one local controller and edge node in each cell. The local controller, in particular, receives and sends data from the near edge node and can also handle and package data on its own. The set of cells is denoted by $Z = \{1, 2, 3, \ldots, z\}$, and the local controller belongs only to its corresponding cell. In addition, $Z'' = Z = \{1, 2, 3, \ldots, z\}$ can be used to represent the group of controllers, with controller $z'$ collecting IoT data from its corresponding cell $z$. The set of controller's energy is denoted by $CE(t) = \{e_1(t), e_2(t), e_3(t), .., e_z(t)\}$, and the computational capacity of the controller is denoted by $CC(t) = \{cc_1(t), cc_2(t), cc_3(t), .., cc_u(t)\}$. Each controller gets various IoT data from local devices while functioning. In this system, an MC is appointed among all controllers to minimize transmission consumption between the controllers and the blockchain network and relieve controllers with insufficient resources. Plus, the MC is in charge of collecting and processing all controllers' IoT data and interacting with the blockchain network. The MC's computational functions can be performed locally and with the aid of the local edge node, which improves performance and saves energy. The following are the computing overhead and energy consumption of the computation functions. If the MC chooses local execution, Equation (1) can be used to calculate the energy consumption $E_{local}$ at time slot $t$ where $q_t$ is the computational complexity of the jobs and $\text{ß}_z$ is the energy consumption coefficient [39], which can be expressed as Equation (2). $c_c$ denotes the controller's computing capability.

$$E_{local}(t) = \text{ß}_z * q_t \tag{1}$$

$$\text{ß}_z = 10^{-27} * (c_c)^2 \tag{2}$$

If the MC chooses to offload the computation jobs, the energy consumption $E_{offload}$ of data transmission paid by the local edge node at time slot $t$ could be represented by Equation (3). $d_i$ denotes the size of the computation task data, $pt$ denotes the MC's

transmission capacity, and $tr_{cd}$ denotes the transmission rate between the local edge node and the controllers [40].

$$E_{offload}(t) = \frac{d_i}{tr_{cd}} * pt \tag{3}$$

Additionally, at time slot $t$, the computation overhead $CO_{offload}$ imposed by the local edge node can be defined by Equation (4). $\xi(t)$ is the fixed charge, and $\Gamma(t)$ is the overhead variable set by the local edge node [41].

$$CO_{offload} = q_t * \Gamma(t) + \xi(t) \tag{4}$$

The MC sends the unverified block to the blockchain system after processing and packaging the data. $E_{blockchain}(t)$ can be used to represent the transmission energy consumption [42]. Additionally, $bn$ denotes the number of blockchain nodes.

$$E_{blockchain}(t) = \frac{d(t)}{tr_{cb}} * pt * bn \tag{5}$$

The transmission rate between the controllers and the blockchain system is expressed by $tr_{cb}$. The overall energy usage $E_{total}(t)$ and the computation overhead $CO_{server}$ paid by the server if the MC performs the computations on its own at time slot t are as follows [40].

$$E_{total}(t) = E_{local}(t) + E_{blockchain}(t) \tag{6}$$

$$CO_{server}(t) = 0 \tag{7}$$

On the other hand, if the MC assigned the jobs to a local edge node, the overall energy usage $E_{total}(t)$ and the computation overhead $CO_{server}(t)$ incurred by the server are being calculated as [43]:

$$E_{total}(t) = E_{offload}(t) + E_{blockchain}(t) \tag{8}$$

$$CO_{server}(t) = CO_{offload} \tag{9}$$

### 3.2.3. The Layer of the Blockchain

The blockchain system has $m$ consensus nodes that are represented by $M = \{1, 2, 3, .., m\}$. They check the MC's transactions and then add the validated block to the blockchain. The blockchain in the proposed method employs a Practical Byzantine Fault Tolerance (PBFT) that could guarantee the system's consistency when the number of misbehaving nodes is less than one-third. Request, pre-prepare, prepare, commit, and respond are the PBFT specific consensus process steps, as shown in Figure 3. The MC first sends the data consensus specification and the unverified block to the blockchain network. The blockchain then assigns a master node randomly to validate the block's relevant information. The chosen master node first verifies the block's signature and Message Authentication Code (MAC). If the above information is correct, the signature and MAC of each transaction will be checked next. Then, the computing cycles are necessary to confirm one signature and generate/verify, and one MAC is assumed to be $\omega$ and $\Phi$, separately. As a result, the request period's computing cycles can be written as Equation (10), assuming $tb$ denotes the transaction batch's total size and $ts$ denotes the average transaction size [42].

$$cc_1(t) = 1 + \frac{tb}{ts} * (\omega + \Phi) \tag{10}$$

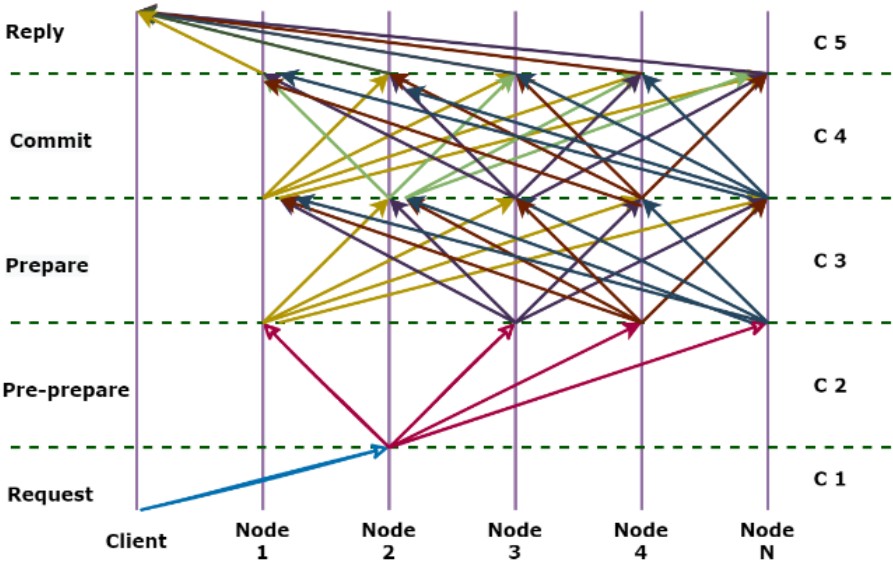

**Figure 3.** PBFT consensus process, step by step shows how to communicate with each other.

The MC then creates a unique MAC and delivers it to each replicated node. The replicated nodes then double-check this MAC and the signature and MAC of each transaction in the block. In this step, the necessary computing cycles at the MC and replicated nodes are expressed by Equation (11) [44].

$$cc_m(t) = (m-1) * \Phi \tag{11}$$

$$cc_r(t) = \Phi + \frac{l \, * \, tb}{ts} * (\omega + \Phi) \tag{12}$$

Additionally, the percentage of right transactions sent by the MC is represented by $l$. The duplicated nodes that have been checked generate and send a single MAC to other nodes. Then, each node receives $(m-1)$ MACs from other nodes and verifies them. The next phase is available if the number of appropriately checked nodes is greater than $2k$ (where $k = m - 1/3$). The MCs and replicated nodes' processing cycles can be interpreted as [45]:

$$cc_{m2}(t) = 2k * \Phi \tag{13}$$

$$cc_{r2}(t) = (m-1+2k) * \Phi \tag{14}$$

If they have received more than $2k$ correct messages, the checked node sends a single MAC message to all other nodes. Meanwhile, each node verifies the MACs of all other nodes. Then, the computing cycles are denoted by Equation (15) [45].

$$cc_3(t) = (m-1+2k) * \Phi \tag{15}$$

Eventually, nodes that have got more than $2k$ valid commit information give the MC one reply response. If there are more than $2k$ correct reply messages, the consensus process is considered complete and efficient. Therefore, the created block is delivered to the blockchain system. There is also block overhead that is proportional to the block size, as follows [46]:

$$CO_{block}(t) = \partial * j(t) \tag{16}$$

Here, $j(t)$ is the block size and $\partial$ is the fixed coefficient to explain the relationship between block overhead and block size. All nodes confirm a large number of signatures and MACs at each stage of the verification process, resulting in a large number of complicated and heavy jobs. Each PBFT consensus mechanism has a time limit $t$ that is proportional to the whole transaction size. The created block will not be modified if the consensus time

surpasses the cap. The efficiency of the consensus is critical. Edge nodes and cloud servers are used to facilitate the verification process of all nodes to deal with these complicated and heavy computation jobs and satisfy the time constraints of the consensus process. Edge and cloud servers have different computational capacities and charge different computation costs. Plus, servers charge extra computational overhead when several consensus processes are needed because of inadequate block size. Since devices share cloud or edge servers' processing resources, it seems difficult to know precisely how much computing power and overhead each server has at any given time. Therefore, the server computing overhead and computation resources are expressed as random variables $\psi$ and $\acute{\eta}$. As $X = \{x_0, x_1, \dots, x_{p-1}\}$ and $Y = \{y_0, y_1, \dots, y_{p-1}\}$, they are partitioned into $q$ and $p$ independent values, respectively. The computation overhead and resources of the servers at time slot $t$ could be represented as $\psi(t)$ and $\acute{\eta}(t)$, respectively. The total computing cycles of one consensus phase could be calculated based on the necessary computing cycles of each consensus stage [45].

$$CC(t) = \left[\frac{tb * (l+1)}{ts} + 1\right] * \omega + \left[\frac{tb * (l+1)}{ts} + 3m + 4k - 2\right] * \Phi, \; if \; tb \leqslant z(t) \quad (17)$$

Whenever the total transaction size exceeds the block size, the overall computing cycles of multiple consensus processes can be calculated as follows (authors considered that there are at most two consensus processes) [47]:

$$CC(t) = \left[\frac{tb * (l+1)}{ts} + 2\right] * \omega + \left[\frac{tb * (l+1)}{ts} + 6m + 8k - 2\right] * \Phi \quad if \; tb > z(t) \quad (18)$$

The consensus time and the consensus process overhead can be calculated as Equation (19) [40].

$$V_n(t) = \begin{cases} \frac{CC(t)}{\psi(t)} + 3b_t & if \; tb \leqslant z(t) \\ \frac{CC(t)}{\psi(t)} + 3b_t + t_i & if \; tb > z(t) \\ \qquad\qquad 1 \end{cases} \quad (19)$$

Additionally, the broadcast delay among nodes is $b_t$, and the block generation interval is $ti$ [48].

$$CO_{bn}(t) = \begin{cases} \psi(t) & if \; tb \leqslant z(t) \\ \qquad 1 \\ 2\psi(t) & if \; tb > z(t) \end{cases} \quad (20)$$

The total overhead $CO_{total}$ can then be expressed as (21).

$$CO_{total} = CO_{server}(t) + CO_{bn}(t) + CO_{block}(t) \quad (21)$$

If the consensus time, $V_n(t)$, is less than the time limit $t_i$, the consensus process is considered active, the created block is appended to the blockchain, and the MC is notified. In this system, Cost($t$) represents the weighted amount of computation overhead and energy consumption, which helps to balance the optimization and reflects the system's overall efficiency. It is referred to as Equation (22). When $\lambda$ is 1, the weighted sum will turn into the cost of energy and vice versa. Besides, we can tune the parameter if one aspect is more important. The numbers were also normalized. There may be a few tasks that cannot be done before the battery dies. If the battery level is equal to zero, the IoT object does not perform the computing task. If a computation task fails, the task failure indicated by $\xi$, is defined as the cost. If failure happens, the purpose of the indicator denoted by $\ddot{\upsilon}(\xi)$ is equal to one, and zero is the opposite. The IoT object's efficiency in selecting the targeted

edge node at time slot $t$ is indicated by $Ef_t$, which is biased on system cost and task failure and is given by Equation (23) [49]:

$$\text{Cost(t)} = \lambda E_{total}(t) + (1 - \lambda)\, CO_{total}\,,\ \ \lambda\,[0,\,1],\ t \in \mathbb{T} \tag{22}$$

$$Ef_t = d_i - \ddot{\text{v}}\xi\,(lb_m = 0) -\ \text{Cost(t)} \tag{23}$$

### 3.2.4. Harvesting of Energy Unit

In the DO$^2$QIEO method, IoT objects that can convert renewable resources to electricity include RF energy harvesters, wind turbines, and photovoltaic modules. It has a battery to help balance the power supply and demand. The stored renewable energy could be used for offloading as well as local computing. The total consumed energy by the IoT object at time slot $t$ is represented by $E_{total}(t)$ with $E_{total}(t) = E_{offload}(t) + E_{blockchain}(t)$ and the amount of harvested energy at time slot $t$ is represented by $\beta(t)$. Additionally, $b(t)$ represents the battery stage at the start of time slot $t$, and it evolves based on Equation (24) [28]:

$$b_{t+1} = \max\{0,\ b_t - E_{total}(t) + \beta(t)\} \tag{24}$$

If the IoT object's energy is inadequate, it will drop the computation operation, i.e., $b_{t+1} = 0$. Additionally, the RF-enabled wireless energy transfer technology in [50], in which the IoT system is operated by a designated wireless power transmitter that produces constant and sustainable microwave energy over the air, is viewed as a special case. Let $W(0,1)$ represent the energy conversion efficiency, $\mu_t$ is the transmitting power, $d_{distance}$ is the distance between the IoT system and the energy transmitter at time slot $t$, $C$ is the cumulative gain of the IoT antenna and the RF energy transmitter antenna at time slot $t$ [45].

$$\beta(t) = W * \mu_t * (d_{distance})^{-\tau} * C \tag{25}$$

The quantity of energy collected at time slot $t$ could be calculated by the IoT system based on the power harvested history and the modeling approach in [51]. The $\beta'$ represents the expected amount of energy produced $(t)$, Additionally, the estimation error for $\beta(t)$ is denoted by $\Delta(t)$ [28].

$$\Delta(t) = \beta(t) - \beta'(t) \tag{26}$$

## 4. Problem Solution

The system model was detailed in-depth in the previous section; in this section, we explain the DO$^2$QIEO method. The system's state and actions are distinct, and the system status changes as operations are performed within a frame. The QL algorithm is a gradual optimization method for selecting difficult actions that converge quickly. Therefore, the method is suggested to direct object behavior to solve the MDP problem. The DO$^2$QIEO method is a hybrid of two algorithms that dynamically perform offline/online functions. The network's status can change from online to offline or vice versa. If the system goes offline, the TL method will be activated and will begin initializing the system's Q-table and other settings as needed. While the network is online, the system enables the PDS technique.

### 4.1. Offline Part

DO$^2$QIEO addresses the problem in this paper due to the large-dimensional and high-dynamic characteristics of the servers and controllers in blockchain-enabled IoT networks, as well as the nature of the offloading problem. While Q-learning can handle formulated MDPs by maximizing reward, it is not excellent. Therefore, if we are using Q-learning, we must calculate and store the corresponding Q-value for each state-action group in a table. In realistic issues, the number of potential states could be in the tens of thousands. If the system stored all of the Q-values in the Q-table, the matrix Q $(s, a)$ would be extremely large. It may be difficult to obtain enough samples to traverse each state, resulting in method

crashes. Besides, rather than calculating Q-values for each state-action pair, we utilize deep Q-learning to estimate Q $(s, a)$; this is the core principle of DO²QIEO. Additionally, the TL is used in the method to initiate the system. To maximize learning outcomes, effective TL is needed. Taking a model that has already been trained in one area and adapting it to a different field has a lot of benefits. Besides, some of the key advantages are less training data, better model generalization, and greater DL usability. TL emphasizes transferring information obtained from addressing an issue to a separate but similar problem, and this method will define the learning parameters. Q-values are initialized using offloading experience in similar contexts, such as multiple outdoor or indoor networks with common edge nodes. The method speeds up learning by reducing random experimentation at the start of the complex computing process. In this method, the controller examines the state $s(t)$, which includes the controllers' energy state, and also the computing resources and overhead state of servers at $t$. Next, the controller chooses and operates, including MC selection, offloading decision, block size, and server selection. In the meantime, the controller is rewarded immediately $r(t)$, and the setting is modified towards the next state $s(t+1)$. The obtained average system efficiency from taking action $a(t)$ in state $s(t)$ could be represented as a Q-function under a given Equation (27). The Q-optimal function's value will then be Equation (28) [52].

$$Q(s, a) = \mathbb{E} \, a[\, Ef_t + \gamma Ef_{t+1}\gamma + \cdots | s_t, a_t] = \mathbb{E}_{s+1}[Ef_t + \gamma Q_\pi(s_{t+1}, a_{t+1}) \, | s_t, a_t] \quad (27)$$

$$Q^* \, (s, a) = \mathbb{E}_{s+1}{}'[\, Ef_t + \gamma \, \max Q \, (s(t+1), a(t+1)) \, | s_t, a_t] \quad (28)$$

Iteration on value and strategy can be used to determine optimal performance as well as the best offloading methods. In each iteration of the learning process, the value of the Q-function is modified as follows, where $\alpha$ is the learning rate.

$$Q(S_t, a_t,) \leftarrow Q(S_t, a_t) + \alpha((Ef_t + \gamma \max Q^*). (S_{t+1}, a_{t+1}) - (S_t, a_t)) \quad (29)$$

On the other hand, the offloading method's conditions are determined by the amount of required computation queuing in the edge devices, which is a continuous value. Finding the best strategy becomes difficult when the state space is divided discretely. Therefore, the proposed traditional Q-learning method cannot be used to solve the MDP. To address this issue, we convert the Q-function to a function approximator, a simple function type that can be used in the optimal action acquisition process. We use a multi-layered neural network [53,54] as a nonlinear approximator to capture complex interactions among different states and actions. We use deep Q-learning techniques to find the best offloading solutions depending on the Q-function estimation. The neural network-based approximator is referred to as a Q-network, where w is the network's set of parameters. Additionally, the Q-function in Equation (27) can be calculated using the Q-network as Q $(S_t, a_t) = Q'(S_t, a_t, w)$. Through iterations, Q' learns to converge to real Q-values. The best offloading policies in each state are extracted from the behavior that results in the greatest efficiency, depending on Q'. At timeslot $t$, the selected action could now be described as $a^* = \text{argmax} \, Q'(S_t, a_t, w)$. The experience replay method is being used in the learning process to maximize learning performance, and the learning experience at each time is preserved in a replay memory. The experience is made up of all observed state transitions and acquired utilities as a result of behavior. $(S_t, a_t; Ef_t; S_{t+1})$ is the experience gained during time slot $t$. A batch of recorded experiences taken at an arbitrary from the replay memory is being utilized as samples in training the parameters of the Q-network throughout Q-learning updates. The training aims to get the gap between Q $(S_t, a_t)$ and Q'$(S_t, a_t, w)$ to be as small as possible. To represent the difference, we use a loss function Equation (30) [23].

$$\text{Loss} \, (w_t) = \mathbb{E} \, a[\frac{1}{2}a(Q'_{\text{tar}} - Q'(S_t, a_t, w))^2] \quad (30)$$

Here $w_t$ is the Q-network parameters at time $t$. $Q'_{tar}$ is a learning objective that represents the function's optimal value in frame $t$ and could be described as [55]:

$$Q'_{tar} = Ef_t + \gamma Q\left(S_t, \text{argmax } Q'(S_{t+1}, a_{t+1}, w_t)\right). \tag{31}$$

To change $w_t$, we use a gradient descent method. The gradient obtained by differentiating *Loss* (*l*) is as follows:

$$\Delta w \ Loss(w_t) = \mathbb{E} \ a\left[\Delta w_t \ Q'(S_t, a_t, w_t)\left(Q'(S_{t+1}, a_{t+1}, w_t) - Q'_{tar}\right)\right] \tag{32}$$

Then, $w_t$ is updated based on Equation (32). Additionally, $\chi$ denotes the scalar stage scale [55].

$$w_t \leftarrow w_t - \chi\Delta w \ Loss(w_t) \tag{33}$$

We use the "greedy policy" to prevent local maximum when balancing exploration and exploitation in the learning experience. To gain better offloading strategies, randomly selected actions are taken with probability $\varepsilon$; otherwise, optimal actions are chosen with probability $1 - \varepsilon$, where $0 < \varepsilon < 1$. The DO²QIEO scheme is depicted in Algorithm 1. The entire procedure can be performed only when the system is turned off. If the system is switched to online mode, the PDS approach will be used to determine the online requirements. The section that follows explains how the system works when it is in online mode. Figure 4 depicts a schematic representation of the DO²QIEO system for IoT offloading computation.

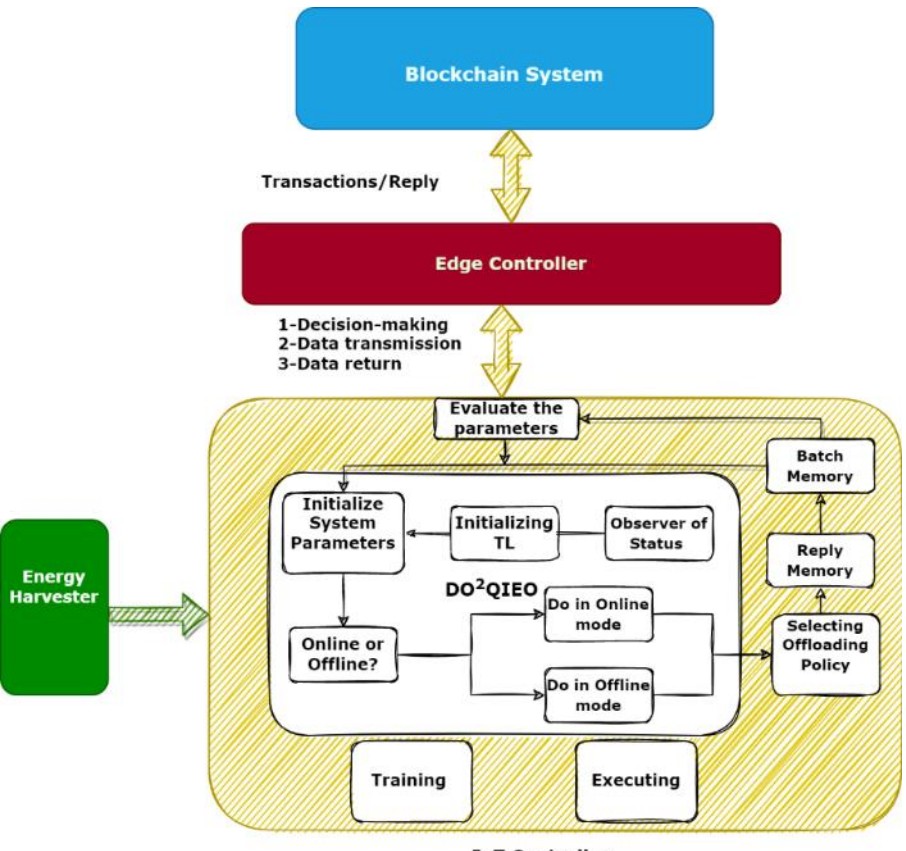

**Figure 4.** The DO²QIEO method is being used to establish a decision-making mechanism.

| | | **Algorithm 1**. DO$^2$QIEO method |
|---|---|---|
| | | **Begin** |
| 1 | | Initialize parameters |
| | | Initialize network parameters |
| 2 | | Initialize action-value function |
| | | Q, and experience replay buffer |
| 3 | | Set Q ← Q based on a TL method trained in Cooja |
| 4 | | t = 1 |
| 5 | | **While** t ≤ n do |
| 6 | | Observe blockchain transaction |
| 7 | | Choose $a_t$= argmax Q′($S_t$, a, w) |
| 8 | | Green energy estimation with Equations (24)–(26) |
| 9 | | Local processing or offload |
| 10 | | **For** system do for every slot |
| 11 | | **If System is online** |
| 12 | | Compute PD state with Equation (34) |
| 13 | | **For** each job, do |
| 14 | | Choose the component of the work that will be offloaded |
| 15 | | Offload or do locally |
| 16 | | **End for** |
| 17 | | **Else** |
| 18 | | **Changes system to the offline manner** |
| 19 | | Derive the next state $S_{t+1}$ and obtain efficiency with Equation (29) |
| 20 | | Save the experience  ′($S_t$, $a_t$, $Ef_t$, $S_{t+1}$) |
| 21 | | **For** each job, do |
| 22 | | Choose the component of the   work that will be offloaded |
| 23 | | Offload or do locally |
| 24 | | **End for** |
| 26 | | **End** |
| 27 | | Calculate the loss function using a batch of samples from the replay memory. |
| 28 | | Check the system status and cost with Equation (35) |
| 29 | | Calculate all needed parameters |
| 30 | | Determine the Loss($w_t$) gradient concerning $w_t$ |
| 31 | | Update $w_t$ with Equations (32) and (33) |
| 32 | | **End While** |

## 4.2. Online Learning-Based on the Decision-Making Process

The situation must be changed once the system is online. Once all of the probability distributions have been predicted, the optimum strategy could be computed offline using standard Bellman equation solving techniques, such as value and strategy iteration. Several of these probabilities are unknown in the situation under consideration, so these techniques are impractical. An online deep Q-learning method has been proposed to generate the best policy on the fly. Thus, our technique employs the PDS mechanism, which uses partially known knowledge about dynamic systems and allows the edge node to incorporate such knowledge into its learning experience to accelerate learning. The DO$^2$QIEO method is based on the PDS approach, which is capable of handling dynamic offloading online. In this section, we will first explain the concept of PDS, which is an essential component of our method. PDS is the intermediate system state that occurs just after the edge node performs

the computational power demand action $a(t)$ but before the $\beta(t)$ is achieved. Since this processing power demand action $a(t)$ has no direct influence on these aspects of the system state, the Post-Decision (PD) of device state for IoT system and tasks remains the same. Moreover, the battery state $b(t)$ is the sole aspect of the system state that could change, which should be noted, nevertheless, that the PD battery state $Pb(t)$ is simply a virtual state and not the actual battery state. Considering the PDS description, we develop the PD value function $PD_{value}(t)$ as shown below [49]:

$$PD_{value}(t) = \Sigma_a P(s'|pS) * \text{Cost(t)} + (s'), \forall s \quad s' \in S \tag{34}$$

If the activity no longer affects the transition $P(s'|pS)$ between PDS and the next state. To distinguish them from their PD counterparts, we refer to s as the "normal" state and Cost (t) as the "normal" value function. The normal value function Cost (t) has a deterministic mapping to the PB value function V (t), which is as follows [49]:

$$\text{Cost (t)} = \min(cost(s, a) + \delta PD_{value}(pS)) \tag{35}$$

The abovementioned equation demonstrates that the normal value function Cost (t) for each time slot is generated from the equivalent PB value function V (t) in the same time slot. The advantages of using the PDS and the PD value function are listed below. To begin, in the standard state-based Bellman's equation, expectations over the prospective state must be completed before minimizing the probable electricity consumption $a(t)$. Carrying out the minimization requires an understanding of these dynamics. In the PDS-based Bellman equations, on the other hand, the expectation process is decoupled from the minimization operation. If we could somehow learn and estimate the PD value function V(t), we could tackle the minimization problem without knowing anything about the system dynamics. Secondly, assuming the energy demand $a(t)$, the PDS breaks down the system dynamics into such an a priori unknown component which evolution is independent of a priori known element, namely the battery state evolution is partially controlled by $a(t)$. The pseudocode for the DO$^2$QIEO is also shown in Algorithm 1.

## 5. Performance Evaluation and Simulation

In this section, we present three subsections: (A) experiment settings and measurements and (B) TL method, and (C) simulation performance. The first discusses the simulation environments, parameters, and hyper-parameters used in the paper. The second discusses the TL technique and its significance. The third explains the results and compares the simulation to other benchmarks.

### 5.1. Experiment Settings and Measurements

Experiments are carried out using both Cooja and TensorFlow. Cooja is built for IoT simulations, especially ad-hoc networks. TensorFlow provides APIs that encourage ML and have a shorter compilation time than other ML frameworks. It is assumed that the battery stage of the IoT objects is within normal distribution. To maximize the IoT object's performance, the object chooses the best offloading policy, with computation tasks offloading to the target edge node. This model has 12 controllers and cells and three computational nodes for the consensus process, incorporating four edge nodes and one private cloud node. In addition, 8 servers are installed in the blockchain mechanism. Besides, Table 3 formalizes the key parameter settings. The compute resource status of the cloud server and edge nodes for the consensus process is categorized into four levels: low, medium, high, and extremely high. Therefore, the compute overhead condition of the cloud server and edge node consensus process is classified as cheap, ordinary, and costly. Moreover, the workload arrival space has been set at = {10 units/s, 20 units/s, ... , 100 units/s}. Additionally, H = {20 ms/unit, 30 ms/unit, ... , 60 ms/unit} is the network congestion space. We build the environment states and extract their associated energy-generating distributions using real-world green energy harvesting traces, including

wind and solar. The time is set at 5 s for two adjacent time slot decisions. The method will infer by experimenting with the optimal offloading policy of the IoT object in the MDP. As expected, the radio channel between IoT objects and edge nodes is stable. An IoT object recharged by an RF energy harvester was simulated to obtain real emergency care advice. The computational assignments are produced at 100 kb/s, with every bit requiring 1000 CPU cycles to complete. The IoT object uses the energy harvesting concept to determine the amount of renewable energy provided by the EH component in this time slot. The EH efficiency is 0.51, and the RF transmitter produces 3 W of corresponding isotropically radiated energy power. Finally, each edge node consumes 150 watts of electricity. Nevertheless, each edge node has a maximum service rate of 20 units/s.

**Table 3.** Parameters Tuning.

| Parameters/Hyper-Parameters | Value | Parameters/Hyper-Parameters | Value |
|---|---|---|---|
| Number of IoT devices | 600 | The block interval | 0:5 s |
| Number of controllers and cells | 12 | Each cell's average transaction batch size | 2 MB |
| Number of users in each cell | 50 | Transaction size on average | 100 MB |
| Number of an edge server | 4 | Controllers' transmission power | 400 mW |
| Blockchain nodes | 8 | Local edge computing power on average | 20 GHz |
| The workload arrival rate | 10, 20, . . . , 100 units/s | Controllers' average computing power | 2 GHz |
| The network congestion | 20, 30, . . . , 60 ms/unit | Noice power | 1 MBytes |
| Two adjacent time slot decisions | 5 s | $\lambda$ | 0.5 |
| Computational assignments generation | 100 kb/s | A | 0.9 |
| Bit required to complete | 1000 CPU cycles | E | 0.15 |
| The EH efficiency | 0.51 | $\Gamma$ | 0.8 |

*5.2. TL Method*

We provide a DO$^2$QIEO approach that leverages the TL method in this section. To maximize learning outcomes, effective TL is required. Taking a model that has already been trained in one sector and applying it to a different one has a lot of benefits. Additionally, some of the primary advantages are less training data, higher model generalization, and enhanced DL usability. As a result, TL focuses on collecting and applying information obtained from overcoming these challenges to a distinct but interrelated issue, and this technique may define learning parameters. Offloading experience in similar contexts, such as many outdoor or indoor networks with typical edge nodes, initializes Q-values.

Additionally, the strategy speeds up learning by reducing random exploration at the start of the dynamic computing process. Almost all of the tasks offloaded to the edge platform were well handled and returned to IoT objects since the edge platform satisfied user expectations. When the edge platform was unavailable or lacked the processing capability to handle new activities, the suggested technique determined that the cloud platform would be used instead. The most serious issue with IoT objects is the issue of power consumption. Data flow and computing are the two factors that determine power. A poor decision may drain a lot of energy. For example, 32-bit processing on an S-RAM chip consumes 5 pico-joules for reading operations, whereas the D-RAM on the exterior of the device consumes 640 pico-joules for the same amount. We must decrease data movements if we are to lower this massive quantity of exponential energy usage. It is here that the correct IoT offloading computation policy comes into play. Therefore, before the network starts, pre-trained data is imported using the TL approach. Although this strategy has some drawbacks, it will shorten network training time and improve network accuracy. Weights and rules can be supplied to the network, and the weights can then be fine-tuned to the data set. Except for specific formulations, TL cannot transfer the representation to the most substantially comparable domains. As previously stated, in any learning but specific problem formulation, there is a lack of reasoning or ability to generate understanding. The network's training time is reduced when this type of TL is used, and the system converges

quickly. A network identical to the fundamental simulation environment is required to extract the target parameters. We tested a similar scenario in which the IoT and edge networks were in the same state. As a result, the targeted and extracted weights closely resembled the core simulation situation.

### 5.3. Result of Simulation

In this experiment, we examine our strategy in various scenarios, taking into account practically every situation that can illustrate how our method is performed. We considered several scenarios in which our method was compared to other benchmarks, with or without modules such as the EH component or the TL. In addition, the method is compared in two benchmark methods in online mode and two more benchmark methods in offline mode. Besides, we are striving to demonstrate how our strategy functions in various scenarios to illustrate how our approach reacts. Therefore, criteria include delay, energy usage, computing overhead, task failure rate, system cost, and device efficiency. The proposed IoT offloading system's network performance is evaluated utilizing four edge devices. The task failure rate has been defined as the proportion of aborted computing tasks. For this reason, we compare the DO$^2$QIEO technique to the four methods listed below. However, we run the simulations in two parts (the system's offline and when the system turns to the online method). At first, we analyze the system in the offline method with two benchmarks (1) Energy-Efficient Dynamic Task Offloading computing (EEDTO) [34] and (2) Blockchain Storage and Computation Offloading (BSCO) [35]. Then, we provided performance evaluation in online mode with two other benchmarks: (1) Online Learning and Control for Green Offloading called (OCGO) [56]. (2) Online Distributed Offloading and Computing (ODOC) [57]. Experiments have revealed that the smallest amount of memory required for DO$^2$QIEO is 100 MB, which IoT nodes can readily backup.

Figure 5a describes the convergence of the DO$^2$QIEO scheme in two modes. According to this Fig, the learning process takes roughly 1893 s to reach the best offloading techniques when TL is enabled and 2400 s when TL is disabled. It is worth mentioning that the DO$^2$QIEO technique in this part of the test is implemented offline in practice. Thus, the expected execution time has minimal bearing on the job offloading app's performance. Additionally, the average IoT offloading object efficiency for diverse cell densities is depicted in Figure 5b. As the number of cells increases, the performance of IoT objects degrades. The utility of the DO$^2$QIEO approach decreases as the number of cells increases. In addition, Figure 5c illustrates the average cost of the proposed approach with and without TL; in this mode, with increasing cells, the proposed method with TL surpasses the proposed method without TL. Therefore, this test demonstrates the significance of the TL, which plays a critical role in this scenario.

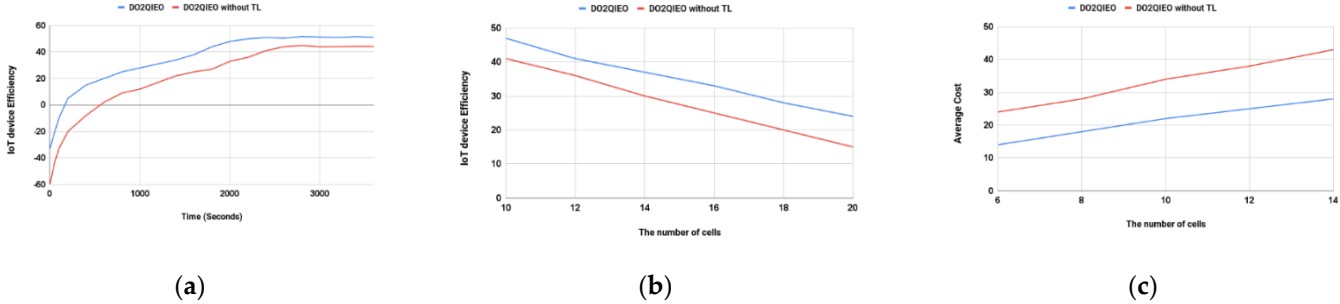

(**a**)    (**b**)    (**c**)

**Figure 5.** The IoT device efficiency in different scenarios, over time slot (**a**), The IoT device efficiency over the number of cells (**b**), and average cost versus cell count using TL and without it (**c**).

Figure 6 describes the relationship between the number of cells and the performance of the network. According to Figure 6a–c, it could be seen that as the number of cells increases, so does the energy consumption, computation overhead, and task failure rate. Furthermore, when compared to other benchmark systems, the DO$^2$QIEO always has a

lower computation overhead. Even though incremental computation assignments use more server resources and larger blocks are created to handle more transactions, the computing overhead rises. Nevertheless, in the DO$^2$QIEO, the acceptable offloading decision, the selection of the cheaper servers, and the quick and efficient modification of block size greatly reduce the additional computation overhead. Assume the number of cells rises from 10 to 14. In that situation, the DO$^2$QIEO technique raises energy consumption, computation overhead, and the rate of IoT task failure by 16.43 percent, 26.36 percent, and 12.5 percent, respectively. With the number of cells equal to 14, the DO$^2$QIEO method outperforms the BSCO by 14.1 percent in terms of energy consumption, 17.81 percent in terms of computation overhead, and 16.42 percent in terms of task failure rate. It outperforms the EEDTO method by 9.42 percent in terms of energy consumption, 8.91 percent in computation overhead, and 12.2 percent in task failure rate. Therefore, when there are more cells, the DO$^2$QIEO performs better. This is because the offloading decision can be dynamically adjusted using the optimal strategy trained by DO$^2$QIEO. The increasing computation tasks are typically offloaded to the local edge node to relieve the MC. The greater the workload, the greater the energy savings from offloading computing. Therefore, energy consumption is effectively reduced.

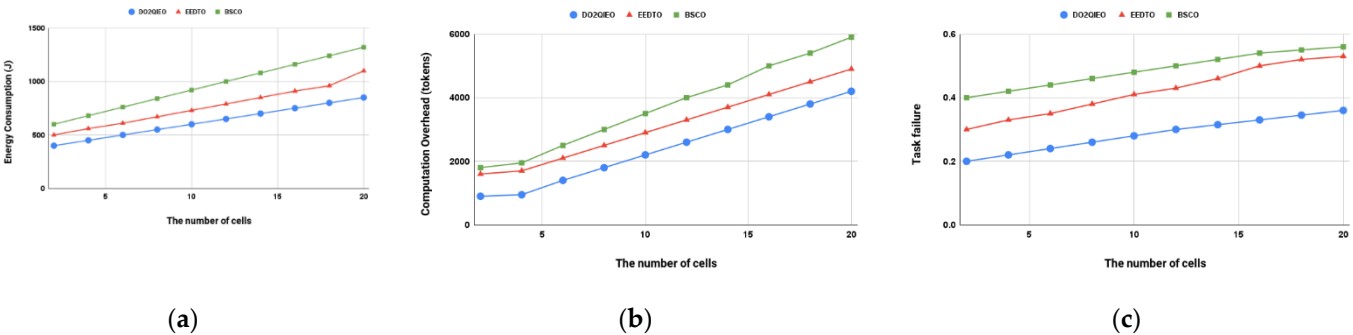

**Figure 6.** Consumption of energy (**a**), Computation overhead (**b**), and IoT Task failure rate (**c**) for the given number of cells.

Figure 7a represents the total network working time following various methods. As shown, the system working time lowers significantly as the quantity of cells increases. The DO$^2$QIEO, on the other hand, outperforms other benchmarks, as evidenced by the extended system working time. A probable explanation is that as the number of cells increases, the computation tasks become more difficult, putting additional strain on the controllers. In addition, the agent prefers to offload tasks rather than execute them locally to save energy. As a result, the system's operating time is prolonged. The system weighted cost is made up of the device's system computing overhead as well as its energy usage. In addition, Figure 7b demonstrates the relation between the system's weighted cost and the number of cells under different schemes. Figure 7 shows that the weighted system cost grows significantly as the number of cells rises in all plans. Because of the combined optimization of the controller selection, block size, offloading choice, and server selection, the DO$^2$QIEO always outperforms alternative baseline methods that just optimize a subset of the optimization elements. Furthermore, due to its capacity to adapt to large-dimensional optimization issues, the DO$^2$QIEO outperforms EEDTO and BSCO in terms of effectiveness and stability.

We proposed a secure framework for integrating edge and blockchain technologies into IoT networks to ensure data security and energy efficiency. The optimization problem was then designed as an MDP by specifying the state, action, and reward function simultaneously, as well as the dynamic features of IoT systems, while taking into account the computing servers and the status of the controllers. Therefore, we developed a platform for dynamic online/offline offloading for various IoT-edge applications that used the TL method to help achieve the best offloading strategy. The EH module was used in our

method to extend the battery's life and improve offloading performance. Results demonstrated superior performance to cutting-edge methods in terms of energy consumption, computational latency, device efficiency, and task failure rate. Therefore, we built three environmental states in our network: E = Low, Medium, and High. E = Low marks the time six pm to six am; E = Medium marks six am to nine am and three pm to six pm, and E = High marks the time nine am to three pm. We generate the environmental states and derive their related renewable power distributions using both wind and solar real-world green power harvesting traces.

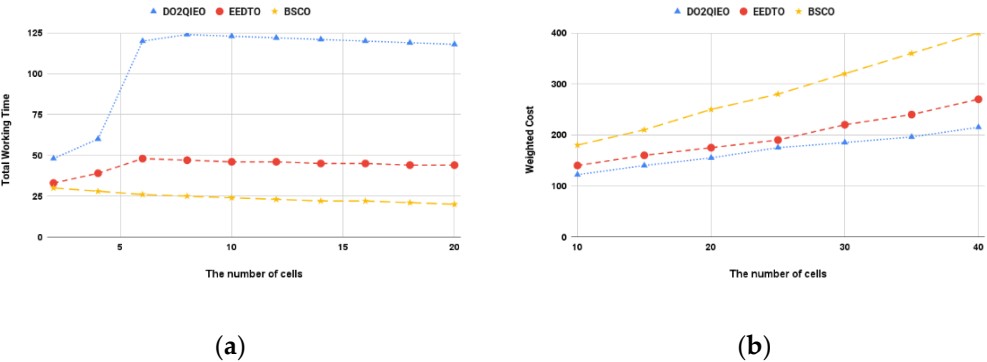

(**a**)                                         (**b**)

**Figure 7.** (**a**) The total working time of the system (**b**) and the system's weighted cost for the given number of cells.

In addition, Figure 8a depicts the energy harvesting path over five days. The delay performance after a 20,000-slot simulation for different maximum numbers of edge nodes in online mode, as estimated automatically by simulation environments, is depicted in Figure 8b. As the quantity of cells rises, the delay for all techniques decreases. Additionally, the DO$^2$QIEO method is faster than all benchmark methods. The suitable offloading choice, the choice of the cheaper servers, and the faster and more effective modification of block size considerably decrease the additional computing overhead in the DO$^2$QIEO in online mode. Suppose the cell count rises from 6 to 10. In that situation, the DO$^2$QIEO technique reduces time by 22%. In terms of latency, the DO$^2$QIEO technique surpasses the OSGO method by 7.6% and the ODOC method by 9.1% with 14 cells. As a result, when there are more cells, the DO$^2$QIEO performs better. This is because the offloading decision can be adjusted dynamically using the optimal approach proposed by DO$^2$QIEO.

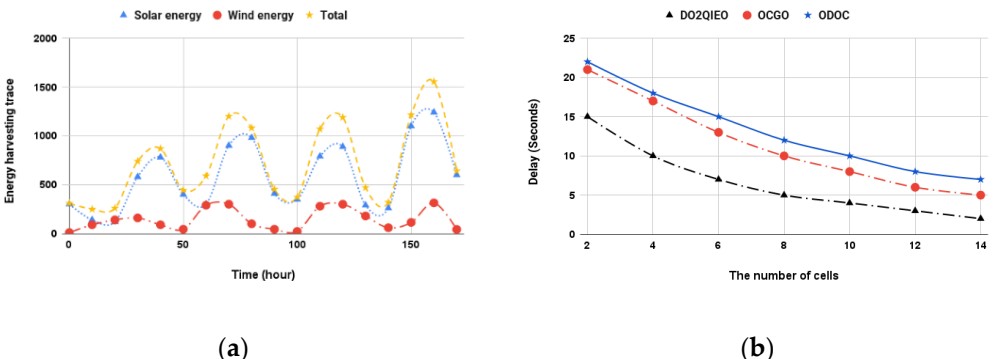

(**a**)                                         (**b**)

**Figure 8.** DO2QIEO technique performance in online mode (**a**) energy harvesting performance over time (**b**) delay of IoT tasks for given number of cells with EH.

Additionally, Figure 9a depicts the task failure rate in online mode after 1000 time slot simulations with different degrees of green energy supply. The study revealed that when the green power supply grows, the task failure rate lowers for all three approaches used in the experiment. This is because with a greater amount of green energy supply, backup power is prevented, and more servers might well be switched on, lowering both backup

power and delay costs. Furthermore, the DO$^2$QIEO has the lowest cost across all stages of the green energy supply. Figure 9b depicts the effect of computing task size on overall computational overhead in online mode ranging from 50 to 300 KB in all approaches; as the size of the calculation task grows, so does the total computational cost. The growing quantity in the ODOC, in particular, is greater than that in the other systems. It is because, unlike previous methods, ODOC performs its functions without the assistance of an energy harvester. Moreover, we could see that the DO$^2$QIEO has a lower total computational overhead than other techniques. It is, in particular, extremely near the result obtained using the OSGO algorithm.

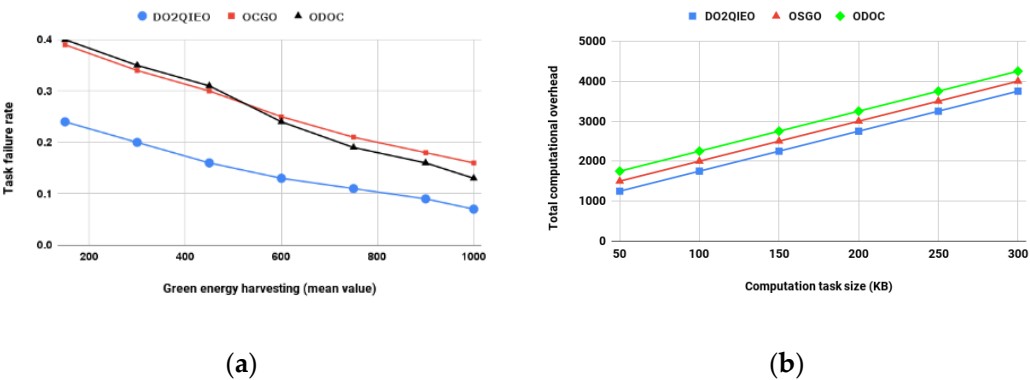

(**a**)                                                   (**b**)

**Figure 9.** (**a**) Task failure rate performance with backup energy harvester module. (**b**) The impact of computing task size on total computational overhead in online mode.

Additionally, we studied two computing situations to demonstrate the efficiency of the system in online mode, which also included local processing and offloading to the edge for IoT data processing. The IoT data are performed locally in the first case. In the second case, we use our settings to offload IoT data for execution on edge computing. The suggested technique was evaluated using a collection of sensor data files ranging in size from 50 kB to 250 kB. Additionally, every IoT data input is run ten times to generate an average result. As seen in Figure 10, we analyzed the outcomes using two performance metrics: processing time (a) and energy consumption (b). In the situation of task offloading to the edge node, the processing time comprises offloading time, execution time, downloading time, and so on. With each IoT data file size, the average processing time of local computation is greater than that of edge computing in Figure 10a. For instance, the offloading system could save up to 38% time when finishing the computation of a 50 kB file and up to 21% time when completing the computation of a 200 kB file, demonstrating the benefits of the offloading method. Because resource-intensive computational processes are offloaded to the edge, IoT data tasks require less energy when done using the offloading strategy, as shown in the results obtained for battery consumption in Figure 10b. Offloading the 150 kB file, for instance, takes below 16% of the energy of local computing. Additionally, when the data size increases, the energy utilization of the offloading system becomes more effective. While running a 150 kB and 200 kB file, for example, offloading the operation to the edge could save 19.33 percent and 21% energy, respectively, when compared to local execution.

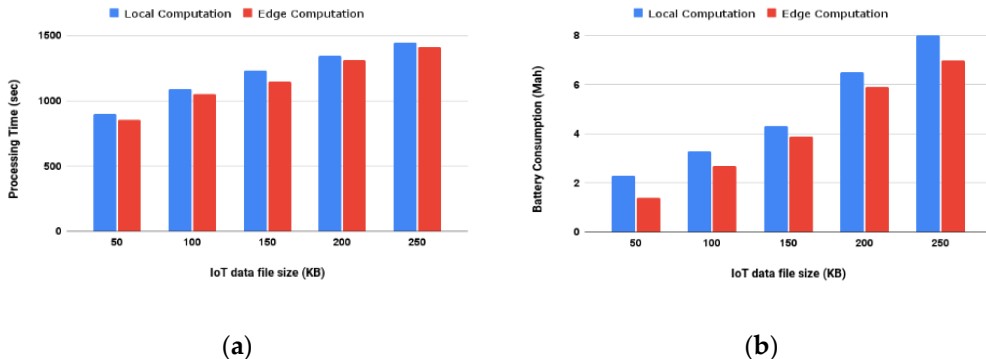

(**a**)             (**b**)

**Figure 10.** Results of online experiments for local and edge computation (**a**) processing time for given IoT data file size (**b**) battery consumption for given IoT data file size.

Finally, the cost compositions of the online and offline algorithms in 20,000-time slots are shown in Figure 11a,b. The online system considerably reduces costs by taking conservative actions at low battery levels and avoiding backup power. On the other hand, offline mode usually causes the battery to enter the inadequate mode, resulting in considerable backup power expenses. Figure 12a shows that increasing the capacity of the edge server has no meaningful effect on the system's average cost. Since more emphasis is placed on energy control during the transmission phase in the DO$^2$QIEO network, the edge side's processing power will not significantly impact the system's average cost. In various scenarios of computation capacity, the suggested algorithm has a lower system-cost than ODOC and OSGO. The simulations are done in circumstances where the channel bandwidth provided in the network extends from 14 to 24 MHz, as shown in Figure 12b. As the system's channel bandwidth allocation rises, the relative advantages of offloading to the edge diminish. The DO$^2$QIEO method, on the other hand, outperforms ODOC and OSGO.

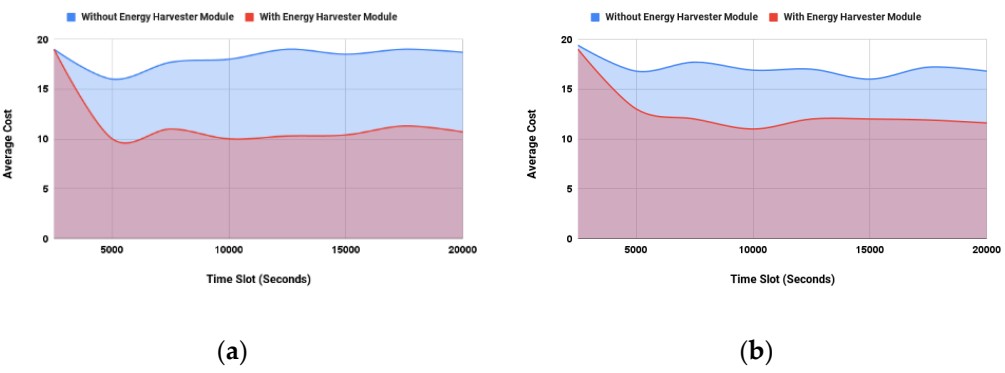

(**a**)             (**b**)

**Figure 11.** The system run-time costs in both online and offline modes. (**a**) Average cost for online mode. (**b**) Average cost for offline mode.

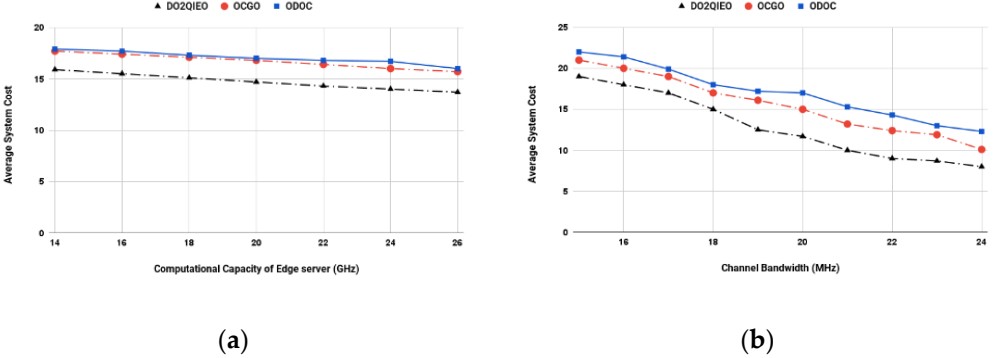

(**a**)             (**b**)

**Figure 12.** Costs of system operation in both online and offline modes. (**a**) Average system cost for given edge server computation; (**b**) Average system cost for given channel bandwidth.

Due to the advantages of scalability, privacy, and security of blockchain, there are several benefits of incorporating blockchain into edge computing. Blockchain is essential for secure data transmission in edge and wireless networks [58]. Within the enormous, distributed edge nodes, blockchain enables dependable connections and management of computing, communications, and storage resources. Blockchain-based service offloading is thought to ensure security when offloading services across numerous edge nodes. Figure 13 compares the results of system latency under various blockchain nodes. The number of blockchain nodes is configured to 2, 4, 6, 8, 10, 12, and 14. Because the number of nodes affects consensus execution time, overall system latency is also influenced. As seen in Figure 13, the DO$^2$QIEO method has the lowest system latency for various blockchain node settings since it produces superior offloading strategies than other methods. BSCO needed the greatest system latency, almost 0.8–0.9 ms more than EEDTO. The rationale for this outcome is that the BSCO has a heavy computational and consensus load. In terms of system latency, the DO$^2$QIEO method has the lowest latency compared to other schemes, implying that our offloading strategy technique may significantly reduce the expected time, particularly in latency-sensitive workloads and multiple node consensus settings.

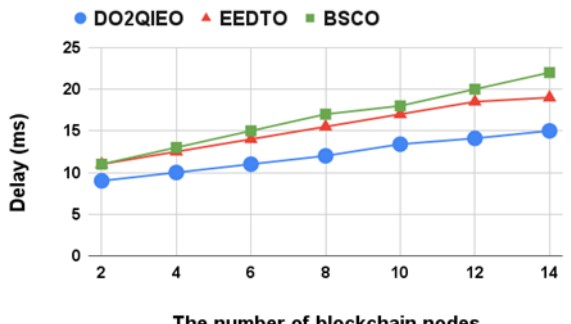

**Figure 13.** System latency in comparison to different blockchain nodes.

## 6. Conclusions and Future Work

This article provides DO$^2$QIEO, a new learning-based technique for offloading in IoT-edge platforms. We used DO$^2$QIEO to solve the formulated problem we modeled as an MDP. This method is applicable in both online and offline environments. Obtaining sufficient samples to explore each state can be difficult, resulting in method crashes. Therefore, rather than calculating Q-values for each state-action pair, we estimate pair state-action utilizing deep Q-learning, which is the key premise of DO$^2$QIEO. The TL system is also employed in the process of initiating the system. In addition, an online deep Q-learning method is presented for the online mode to generate the optimum policy on the fly. In online mode, our solution applies the PDS mechanism, which makes use of partially known knowledge about dynamic systems and allows the edge node to incorporate such knowledge into its learning experience to accelerate learning. Moreover, edge computing is employed in this system to reduce reaction time and conserve energy. Our proposed technique implies that RF energy harvesters, wind turbines, and solar modules are examples of sources linked to IoT devices that can convert renewable resources to power to extend battery life. In addition, there are various drawbacks in present blockchain-enabled IoT systems, such as inadequate efficiency of the blockchain's consensus mechanism, significant computing overhead of network systems, and unacceptably high energy consumption for computation jobs. To address the concerns and challenges, we have integrated an edge system into blockchain-enabled IoT networks to boost IoT objects' computing capability and the consensus process's competence. Compared to traditional online DL techniques, the suggested technique substantially enhances high processing and is managed efficiently. The circumstances for both local processing and fully offloading are supplied, and the performance of the proposed system after convergence is tested under various standard situations. A comparison of DO$^2$QIEO and benchmarks revealed that the DO$^2$QIEO out-

performed in reduced energy consumption, system cost, lower computational overhead, shorter delay, and lower task failure rate.

For future work, the Internet-of-Behavior (IoB) is thought to be the next generation of the IoT. The IoB is based on IoT and results in dynamic adaption and behavior creation. Due to its networked nature, data analytics could be used to adapt and manipulate behavior rapidly. Our next project is online dynamic offloading in the IoB environment. Furthermore, task reliability is a critical scenario due to the numerous network components. It is usually true that the service provider must ensure the dependability of the edge servers. As we all know, determining user-side reliability is difficult. Future researchers can also use 5G/6G technologies to tackle this problem. Additionally, using other methods such as a recurrent neural network [59,60], fuzzy sets [61], recursive neural net (RvNN) [62], Elman neural networks (ENN) [63,64], and temporal convolution network (TCN) [65] to solve this problem can be investigated in the future. Additionally, the efficient mutation operator and Gaussian process regression [66–68] can increase the efficiency of the proposed method. Finally, considering other parameters such as multistability [69] must be investigated in future work.

**Author Contributions:** Conceptualization, A.H., methodology, A.H., and M.A.J.J. and N.J.N.; software, A.H.; validation, M.A.J.J., N.J.N. and S.A.; investigation, A.H., M.A.J.J., N.J.N. and S.A.; writing—original draft preparation A.H.; writing—review and editing, N.J.N.; supervision, M.A.J.J. and N.J.N. All authors have read and agreed to the published version of the manuscript.

**Funding:** This research received no external funding.

**Institutional Review Board Statement:** Not applicable.

**Informed Consent Statement:** Not applicable.

**Data Availability Statement:** All data are reported in the paper.

**Conflicts of Interest:** The authors declare no competing interests.

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
