# Peer review of "Deep Q-Learning Technique for Offloading Offline/Online Computation in Blockchain-Enabled Green IoT-Edge Scenarios"

_applsci, doi:10.3390/app12168232_

Round 1

Reviewer 1 Report

Abstract: Please define what "offloading" is before talking about previous work which does not sufficiently deal with offloading. 

Introduction:

Line 44-same as above point, please define briefly what "offloading" is, before talking in detail about it. 

Line 68/69: what are these "suitable decisions" you are talking about. Please explain with an example

Line 85-87: Please describe what happens during transitions from offline to online with a simple and clear example

Line 87: Rephrase-"the significant contributions in this paper are"...

Line 101-103: Rephrase-"the paper is organnized as follows"

You are talking about the contributions in this paper, are all of these novel what you have listed from lines 89 onwards?

Related work: 

In table 1- you could align the "main idea" column better so that they occupy more horizontal space and not vertically dragged as it is, so that readability is better

Line 180: Please quantify and/or qualify what you mean by "not properly" investigated. Such statements do not make any sense scientifically, especially in related work section

Line 187-188: There is a bold font type for "system model and problem statements"- is it part of a separate section or subsection?

Line 191: Until now, the idea of "offloading" has not even been briefly stated. 

Line 211 -above this text there is incomplete line

Line 211-241: The problem formulation is a huge chunk of text. It can be made better readable and understandable by splitting it into small paragraphs and/or listing the salient points

In Table 3: Its not clear what you want to convey. please distinguish between the parameters and hyper parameters and write clearly what you want to convey in this table

Lines 583-588: please write in detail about "numerous scenarios" and "benchmarks". General remarks as the ones mentioned in these lines carry no significance in a scientific paper.

Lines 652-681: Please list the contirbutions of the study in bullet points and write clearly the significant results. "plus" word must be avoided and replaced with a more scientific usage

The subsection "C-Result of simulation" must be entirely rewritten to clearly group the results and contributions. A clear qualitative and quantitative analysis and comparison of results is expected (nevertheless not available)

Author Response

Dear reader, please examine the response file. We hope our response meets your expectations.

Reviewer 2 Report

In this paper, the authors offer a new deep Q-learning approach to address the IoT-edge offloading enabled blockchain problem using the Markov Decision Process (MDP). This system can be used in both online and offline settings while maintaining privacy and security. The proposed method employs the Post Decision State (PDS) mechanism in online mode. Also, the authors integrate edge/cloud platforms into IoT blockchain-enabled networks to encourage the computational potential of IoT devices.

This research is meaningful. However, please find below some comments to make the paper stronger than its initial submitted version:

1.        Authors should use Arabic numerals instead of Roman numerals when introducing the structure of the paper at the end of Section 1.

2.        Related work should be a separate section to make the structure of the paper clearer.

3.        Please use higher-definition figures for all the figures in this manuscript. Many of the figures (e.g., Fig.2) in the current version are a little blurry.

4.        The method proposed in the paper needs a more detailed introduction. The current version lacks some details of the design.

5.        The evaluation work on the use of blockchain technology in the paper needs to be enhanced.

6.        The results of the evaluation require more detailed analysis.

Author Response

Dear reviewer, please examine the response file. We hope our response meets your expectations.

Reviewer 3 Report

The paper is interesting, well-written and easy to follow.

The approach is sound and validated by experiments.

I suggest only to better format table 1 which is quite unreadable.

Author Response

(The authors gave the same response as above.)

Round 2

Reviewer 2 Report

The authors has addressed all my comments.